# Power and Subjectification at the Edge of Social Media Interfaces in the Aftermath of the Jallikattu Protest

Deepak Prince 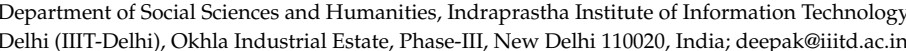

Department of Social Sciences and Humanities, Indraprastha Institute of Information Technology Delhi (IIIT-Delhi), Okhla Industrial Estate, Phase-III, New Delhi 110020, India; deepak@iiitd.ac.in

**Abstract:** In January 2017, millions of people occupied various public places across the southern Indian state of Tamil Nadu, protesting the Supreme Court's ban on *Jallikattu*, a bull-wrangling contest considered central to Tamil identity. Social media was thought to have triggered this 'leaderless' protest. Seven days in, a police crackdown splintered the protest's seemingly unified front. Academic commentators have argued that social media present radical possibilities, 'short-circuiting' older forms of broadcast media, which had already been colonized by the state. Taking as discursive sites two videos, one of them posted by a popular Facebook group and another by a YouTube channel centred around Dalit issues, I argue that an a priori claim of new media having a lesser or greater potential to resist colonization is largely untenable. The possibility of such resistance is contingent on the micropolitics of contestation within concrete, localized sites. I analyse narratives of loss and rage on two different social media spaces, elicited from a fishing community near one of the protest sites, after their homes were attacked and their local market had been burnt down by the police. By focusing on tactics of interviewing, I demonstrate that, in the span of a week, the same technological platform credited with sparking the protests that brought the Tamils together as one, now constitutes the limits of the formation of radical subjectivity, as Tamil society finds itself fractured once again.

**Keywords:** social media; protest; Tamil Nadu; digital media; networks; caste; power; Jallikattu

## 1. Introduction

In January 2017, over a million people occupied a number of public spaces in different parts of the southern Indian state of Tamil Nadu, protesting the ban of Jallikattu, a traditional bull-wrangling sport. This was one of the largest political assemblies in post-independence Tamil Nadu. Social media was thought to have played a definitive role in sparking this protest, both in public discourse and in scholarly commentary (Poorvaja 2017; Kalaiyarasan A 2017; Ravindran 2020). This article focuses on the aftermath of the protest and seeks to examine the micropolitical limits of contestation in the events surrounding the final days of the protest, when a police crackdown ruptured the seemingly united front presented by the Tamils. My objective is to produce a critique of views that treat 'social media' as a self-evident category that is thought to encode characteristic power effects through 'affordances'; Shaw (2017, p. 598), for instance, identifies a 'dominant/hegemonic use' of social media as those affordances that designers intend for the users. He opposes this to resistive use: those affordances that users imagine beyond what the designers had planned for. Within this scheme, power is a relationship between user and designer, conceived as self-evident whole categories. I propose, in this article, to displace the analysis of power and resistance from the opposition user–designer. My approach, extending Foucault (1995), is to examine differential power effects at the edge of interfaces that constitute different kinds of subjects. I show, through two contrasting case studies of videos published on social media platforms, how targets of police violence are produced on the one hand, as political subjects speaking the language of force and justice and, on the other hand, are produced as depoliticized victims in need of aid, through two different tactics of interviewing. I limit

my analysis in this article to power effects at the edge of social media interfaces, and in a longer work in progress, I situate this problematique in relation to the extended matrix of the politics of the state and central government in post-colonial Tamil Nadu.

Social media activism is framed broadly within two contrasting lines, both in public discourse and in academic scholarship. One takes the view that activism on social media amounts to little more than 'clicktivism', a position of inferiority in relation to 'on the ground' substantial forms of protest. For instance, Chattopadhyay (2011, p. 65) writes that a transition must be achieved from online communication of 'token protest, resistance and support' to an offline space of 'tangible, material confrontation' in order to avoid falling into clicktivism. New media spaces online, writes Chattopadhyay, 'serve as a surrogate socio-political space', a preparatory or developmental space that is 'prior to the deployment of the movement in real politic'. According to Bonilla and Rosa (2015, p. 10), online spaces '[may] represent fleeting moments of awareness, quickly replaced by the customary innocuousness of social media pleasantries'. But, they suggest, such spaces are 'also inherently aggregative', allowing for an accumulated or cumulative build-up of affect, which at some point or the other, may possibly express itself explosively as political practice. The other view sees social media as a vital catalyst, a trigger, which, when pulled, explodes into a mass protest (Parikka 2007; Postill 2012).

In his analysis of the Jallikattu protests in Tamil Nadu, Kalaiyarasan A (2017) suggests that 'pent-up anger' against the central government's policies that were seen as anti-Tamil, with attempts by Hindu organizations to make inroads into Tamil politics, global corporations that were seen to extract resources from Tamil Nadu for their profits, and the perceived ineptitude of the state government, all combined, find their 'trigger' in the Jallikattu ban, exploding as mass protests. Ravindran (2020, p. 25) suggests that the event was 'misread by many outside Tamil Nadu as an expression of solidarity against the ban on Jallikattu'. He is also of the opinion that the Jallikattu ban, coinciding with the lack of a strong government after the death of Jayalalitha, the sitting Chief Minister of Tamil Nadu, 'triggered [the Tamil's] pent up anger and anguish on a range of issues'. In both these accounts, 'social media' is seen to provide a 'growth trajectory' (Ravindran 2020, p. 26), a 'mobilization' (Kalaiyarasan A 2017) for a set of trigger-factors.

Cody (2020a, 2020b) situates the problem in relation to transformations in the form of political sovereignty in Tamil Nadu, which coincides with a turbulence caused in the sphere of representational media (cinema and TV news), following the arrival of new media technologies such as social media. Cody rehearses this history thus: since India's independence, politics in Tamil Nadu has been deeply intertwined with the Tamil film industry, as leaders of the major political parties were formerly movie stars and script writers. With the liberalisation of the economy, the two major political parties in the state turned to television, each owning a satellite TV network broadcasting propaganda, serial soaps, and programming from the world of cinema. The most crucial element was their control of news. Each news channel would engage in a symmetrically opposed critique of the policies and practices of the other party. This landscape, according to Cody, is 'short-circuited' by the arrival of 24-h news channels that are not party affiliated and even more so with the coming of social media. Content that is widely circulated across social media did not have to rely on the 'established circuit' of cinematic media or mainstream news media, thereby 'short circuiting', i.e., bypassing the news media circuits. He suggests further that news media finds content to air on their channels by drawing on social media circulations, giving these circulations a 'meta-publicity', that is, a reflexive commentary that recirculates the social media content on news media networks. That is to say, social media intervenes in the field of political power by 'short-circuiting' channels of power thus far colonised by the dominant political parties, offering new political possibilities. In his characterisation of the Jallikattu protest, Cody does not explicitly invoke the notion of a 'trigger' and 'explosion', but these metaphors, in my view, reappear displaced and translated as 'magnetization of affect' around the figure of the bull, as the interplay of 'short-circuiting' and 'meta-publicity', a logic of explosive contagion.

I argue that these views, in treating social media as a totality (either as impotent 'clicktivism' or as contagious 'trigger') foreclose their analysis to the microphysics of power (Foucault 1995) that detotalises the social media landscape. My focus is the edge of the screen and the social media interface, a site that is neither online nor offline, but at the edge. I situate the Jallikattu as a key symbol in Tamil self-conceptions of identity and then provide some background to the political context of the days preceding the protest. I then outline the occupation of Chennai's (the capital of Tamil Nadu) Marina Beach, a long shoreline that is a prominent city landmark. I then detail the ending of the protest at the hands of the police. The thrust of my analysis will draw on two case studies in the aftermath of the protest. Both cases involve videos made in areas affected by police violence. I trace the differential relations of power that frame subjectivity in either case and locate their position in the post-protest discourse to delineate the microphysics of power whose contours emerge through the fractures in the short-lived Tamil political community of resistance. A project of decolonising media must trace these microphysical lines of force.

## 2. Pongal, Jallikattu and Tamilness

The harvest festival of *Pongal,* celebrated on the first day of the Tamil month of *thye*, (typically during the third week of January) is the most important calendric event for the Tamils, both in India and elsewhere, marking thanks for a successful harvest. In contemporary times, when the farm life no longer dominates the lives of many Tamils, Pongal is the time of the release of new Tamil films. Pongal is also a time for television specials, as small-screen celebrities relinquish, if only for a day, their usual roles in family tele-dramas and, instead, engage in banter on talk-shows and compete in games.

Agrarian life or not, in many parts of Tamil Nadu, the Jallikattu is the flagship event of the festival. *Jallikattu* or *Yaeru Thazhuvuthal* (embracing the bull) is something of a metonym for Pongal. The event involves large native bulls, specially bred, fed and trained with care by their owners, unleashed into an arena of contestants—young men, who, like the bulls, have trained for the event. The challenge for the contestants is to embrace the onrushing bull by its prominent hump and to hang on for a certain distance without letting go, at the risk of being trampled or gored by the massive horns of the bull. In the past, the young men who win the Jallikattu were betrothed to the daughters of prominent men. Bulls that eluded contestants brought glory to their breeders and would retire as studs.

Jallikattu is central to Tamil practices of masculinity. The rushing bull, in songs and in colloquial speak, is a metaphor for a fearless and virile young man specifically among the land-owning castes. In recent times, the Jallikattu from prominent venues, such as Alanganallur in Madurai, has been televised. Winners often receive prizes—motor bikes, cars, consumer electronics and cash. If a bull eludes capture, the animal's owner wins the reward. The prizes are sponsored by local bigs, and organizing the event is a way for those with political ambitions to win the favour of people in the area.

Even though agriculture is no longer directly part of the lives of most Tamils, Jallikattu and the Pongal festival are very much key symbols of Tamilness. In 2004, a case was registered by People for the Ethical Treatment of Animals (PETA), claiming that bulls were harmed in the conduct of Jallikattu through the administration of pain and discomfort, in order to induce ferocity in them. Since then, the status of the sport has been mired in legal controversy. The state government has sought out workarounds almost every year since.

In 2009, the state legislative assembly passed an act regulating the conduct of the sport. The Supreme Court granted permission for the event in 2010 under the supervision of district collectors so long as certain guidelines were followed. But, in 2014, the Supreme Court struck down the Tamil Nadu legislative assembly's act, outrightly banning Jallikattu. The court, in its ruling, also opined that the physiological constitution of the bull was ill-suited for the sport and that it was inherently cruel to the animal. In January 2016, the central government issued a notification permitting Jallikattu to be conducted. This notification, however, was challenged in court by PETA and the Animal Welfare Board of India. The Supreme Court stayed the central government's notification and upheld the ban.

Since the time of these legal challenges, the promise of conducting Jallikattu hassle-free was central to electoral mandates: 'You give us the vote, we will give you Jallikattu'. The circumstances leading up to the Pongal of 2017, however, were different.

The state was facing the worst drought in over a century. As crops charred under the sun, farmers, sunk deep in debt, were in despair. Farmers' suicides were on the rise. Only the previous year, heavy floods had taken an immense toll on Chennai, Tamil Nadu's capital, and other areas on the south-eastern coast. And, in December 2016, a powerful cyclone made landfall. Seaside settlements were washed away, and many houses were severely damaged. At least twenty-four people were reported dead. Over a hundred thousand trees were uprooted in Chennai alone. On the fifth of December 2016, J. Jayalalithaa—six-time Chief Minister (CM) of Tamil Nadu and leader of one of the two major political parties in the state, the Anna Dravida Munnetra Kazhagam (ADMK)—was declared dead, having spent the last seventy-five days in hospital. It was the end of an era in Tamil politics, for Jayalalitha was a figure of immense authority who inspired fear amongst the members of her party and commanded the grudging respect of her political opponents. Her public funeral saw millions gather in Chennai, as many Tamils grieved the loss of their leader (BBC 2016). O. Panneerselvam was sworn in as the new Chief Minister of Tamil Nadu. He was known as a trusted loyalist and as Jayalalithaa's puppet in some quarters. Jayalalitha was a tough act to follow, and Panneerselvam had to contend with various intrigues within the party as factions sought to promote candidates of their choosing.

It is in the shadow of these events that the latest ban on Jallikattu surfaced. Sporadic protests, demanding the ban be revoked, (a regular feature of the last decade), started once again in December 2016. The Dravida Munnetra Kazhagam (DMK), which was the major opposition party, the Naam Thamizhar (We, the Tamils) party, the Communist party of India and students in various colleges organized protests in many locations, including certain key villages near Madurai. Celebrities from cinema endorsed the protests too.

The issue boiled down to this: on what basis does PETA or the Supreme Court get to decide how 'we the Tamils' play with our beloved bulls, play as we have been playing for decades, centuries, millennia? The ban was seen as an attack on native bulls, the native genetic stock, native practices of human–bull relatedness, and most of all, on native Tamils. It was alleged that PETA and the other organizations opposed to Jallikattu were motivated by shadowy economic interests from abroad that sought to replace and homogenize the genetic pool of cattle, a veritable colonization of the cattle gene pool. Some protests called for the animal ethics organization PETA to be banned, for its investments to be investigated to understand their true motivations (The Hindu 2017).

The police broke up some of these protests, and some arrests and preventive detentions were made. Meanwhile, in certain villages near Madurai, Jallikattu was conducted, disregarding the Supreme Court's ruling on the matter.

## 3. Protests in Chennai's Marina Beach

In December 2016, the Care and Welfare Foundation (CWF), a Chennai-based NGO, in conjunction with a social media marketing agency known for running the extremely popular Facebook page 'Chennai Memes', launched a campaign to 'save Jallikattu'. These organizations had risen to prominence following their active participation in relief efforts during the floods that ravaged the city a year before. The Chennai Memes Facebook page had been creating and sharing memes by the dozen every day on the subject of Jallikattu. They invited their subscribers to upload a photo of themselves with a banner declaring their support for Jallikattu and asked them to share it online with a particular set of hashtags. They were not the only ones, as several other Facebook pages produced memes and images on Jallikattu and its relation to Tamilness and shared extensively on Facebook and WhatsApp.

Towards the end of December, CWF and Chennai Memes released a poster, an invitation for an event scheduled for the 8th of January[1]. It added: 'For all those who asked, "What's the point of holding a banner at home, clicking a photo of yourself holding it up,

sharing it online and getting a hashtag trending?", this is a call for you. Please come to Marina on Jan 8th Morning at 7:00 AM Sharp to see the real Tamizhans who has the feel for our tradition in blood by birth!. . . We are not just online poralees [protesters]'. On Sunday, January 8th, over twenty thousand people turned up for the walkathon at Chennai's Marina Beach. There was even a bull, all decked up, much to the thrill of some of the children who had come for the walkathon. And there were drums. It was a celebratory walkathon. Many selfies were clicked and shared. News channels had a field day with the event as the images of the walkathon were beamed repeatedly on multiple 24-h news channels. Two days after the walkathon, CWF published a photo[2] on their Facebook page, showing a group of their volunteers gathered around the smiling CM O. Panneerselvam. They had asked for a permanent solution to the Jallikattu problem, and the CM had expressed his interest in seeing the matter resolved. This feeling of victory, however, was short-lived.

On the 12th of January, the Supreme Court rejected pleas requesting an urgent judgement on the case filed by PETA and the Animal Welfare Board of India challenging the centre's 2016 notification permitting the conduct of Jallikattu. A few villages near Madurai held Jallikattu events on Pongal day, openly defying the Supreme Court ban. Alanganallur, a key site for Jallikattu, saw a series of protests organized by political parties, by student organizations, and by other interest groups. On the 15th of January, protesters at Alanganallur staged a sit-in near the 'vadivasal'—the narrow gateway through which the Jallikattu bulls would storm out of. Around two hundred of them, mostly students from the surrounding districts, stayed there through the night, cooking dinner for themselves and feasting together. They declared they would not return to their homes until the vadivasal gateway opened for the bulls. Early the next morning (16th January), a large police contingent removed them forcibly from the site and detained them in a marriage hall a few kilometres away. The site, however, did not remain unoccupied for long. Students and village residents attempted to enter the temple near the vadivasal, along with their bulls. They were repelled by the police who wielded batons now. Bulls were let loose, and it caused some tension. The crowd scattered when the police descended on the protestors with their batons.

Images of protestors being arrested at Alanganallur made the rounds on news channels in Tamil Nadu and in social media. On the morning of the 17th of January, a small group of people assembled at Marina Beach, protesting the arrests in Alanganallur. A few of them had participated in and helped organize the walkathon I spoke of earlier. They were also raising slogans demanding that the ban on Jallikattu be lifted. They were not members of any single organization or political party. Throughout the day, their numbers grew as word spread on social media and as the different 24-h news channels gave the protest quite the coverage, featuring interviews with protesters in English and in Tamil. The protesters declared that they were not leaving until their demands were met or were addressed directly to the CM. They remained on the beach through the night.

While the first day of occupation had seen an ominous build-up of police forces in the Marina area, the 18th of January saw little police action. The police had cordoned off a section of the road to let vehicles pass, but the protesters were left unmolested. News channel coverage was dominated by the scenes at Marina, even as memes and images flooded social media pages and WhatsApp conversations. People passing by the protest site on the marina sometimes stopped their vehicles, folded in by the spectacle, sat among them for a while and then went on their way, only to return in the evening.

On the 19th, the numbers grew to the order of hundreds of thousands, as men, women and children, IT professionals, students, others who had migrated to the city for work, all gathered at Marina Beach. 'The peaceful protests at Marina' was the dominant subject on news media, to say nothing of social media. A large platform was set up on the marina shore, public announcement systems had arrived, and news vans set up shop at the beach, giving the site 24-h coverage. Apart from this first platform illuminated by powerful lights set up by the TV stations, a few more platforms equipped with a mic and speakers were set up, each at less than a hundred meters from the next one, stretching across the shoreline.

These platforms served as public spaces where the Tamils addressed fellow Tamils, and discussions extended well beyond the legal complexities of the Jallikattu case.

The unifying slogan of the protest was to revoke the ban on Jallikattu. PETA, the animal rights organization, drew most of the ire from the protestors. It was claimed that the organization painted the Tamils as savages who mistreat their beasts, an imagination that bore echoes of old colonial anthropologies of the native, lacking in enlightenment values. And, to further their claims, PETA was seen as using the institution of the Supreme Court, part of the colonial legacy in India, to intervene in a cultural practice of the Tamils that dated back to a deep past. Alongside PETA, other global organizations such as Pepsi and Coca Cola, were targeted for destroying local water bodies, part of the community commons, and selling the people sugared water in return. Politicians claimed the protestors were allied to forces that were against the people, and the protest site was declared strictly off-limits to the various members of the legislative assembly and those from the opposition parties as well. The other major cultural figures in Tamil imagination, the movie stars, were also prevented from entering the protest sites. 'Support us from the outside if you want, but you will not enter our space. It is we who will speak here', was the message. That said, the protest brought to the fore its own list of mini-celebrities, who were seen as existing in a relationship of horizontality with the protestors—RJ Balaji, a popular radio jockey, Adhi, a hip-hop artist, and Karthikeyan Sivasenapathy, a farmer, bull-breeder and long-time campaigner for the Jallikattu cause.

With the protest now entering its sixth day, the Chief Minister (CM) of Tamil Nadu, O. Panneerselvam, announced that an ordinance was being drafted, which would allow for Jallikattu to be held. With the passing of the ordinance, many of the prominent faces of the protest announced that the objective was met and that it was 'time to go home'. On the morning of the 7th day of the protest, policemen entered the beach and demanded the protestors leave. The protestors resisted, saying that they wanted to see proof of the ordinance's passing, some insisting that they would not leave until a permanent resolution to the problem was achieved, seeing the ordinance as another ad hoc solution only for the issue to bubble up again. Still others felt that the protest was more than just about the Jallikattu, and they wanted to stand ground, holding the radical space that had been created.

The batons were soon out and protestors, many of them students, were sent scurrying into the by-lanes of the settlements bordering the beach, where the fishermen lived. Policemen followed them into the fishing hamlets, breaking down doors of houses, assaulting the fisher-folk for sheltering the protestors. Witnesses from the area alleged that some of the police personnel set the fish market ablaze with "a white powder", opined to be white phosphorous. Hundreds of arrests were made all over the city, as protests broke out in other parts when news broke out of the dispersal by force at Marina. In the days following the crackdown, the fishing hamlets of Nadukkuppam, Ayothikkuppam and Rutherpuram wore an eerie pallor, as males between the ages of sixteen and about sixty were not to be seen. They had either been arrested or were hiding away, fearing arrest.

The official line was that the protestors had turned violent, that the until-then peaceful protests had been infiltrated by 'anti-social elements' (PTI 2017). Soon, images surfaced online depicting uniformed policemen setting fire to vehicles and to huts. The beach, by this time, had been cleared out. A curfew was declared in the city and in other parts of Tamil Nadu. The protest was over.

Among the many who were arrested on the 23rd, the CM announced at a press meet that 36 students were soon to be released. The others who remained arrested escaped from the conversations on news channels and on most screen spaces of social media, slipping under the blanket of 'anti-socials'. There were vague promises that, if indeed policemen had violated proper conduct, appropriate action would be taken (ANI 2017). Save for a few activist groups and human rights lawyers, most of the Tamils either remained ignorant of their plight or believed that what began as a good people's movement, had fallen apart under the bad influence of some rotten apples who now received their just deserts.

The 'success' of the protest was that it achieved the passing of the ordinance that created a legally valid means of bypassing the ban on Jallikattu. The protest's aftermath, that of police violence, and the alleged violence of 'mischief makers', did not long dominate either the television channels or the social media platforms. Singh (2017) sees the ordinance as a win against a form of colonial mentality that saw Jallikattu as culturally inferior to the elite sport of polo, which, once again, involved the participation of an animal. For Kalaiyarasan A (2017), the protest represented a 'Tamil Spring' with social media, serving as a vital trigger in this politicisation. Cody (2020b) indexes the role of WhatsApp, both in the publicising of the protest and in the aftermath, as a form of citizen sousveillance, 'well outside of the official trajectories of circulation'. Mainstream media, to make sense of this, had to turn back into the field of social media to draw its content, thereby acknowledging that social media had successfully short-circuited established forms of circulation. Cody cites the use of WhatsApp by protestors coordinating between themselves to avoid clashes with the riot police. These views, of the intrinsic decolonizing potential of social media's affordance to sousveil thanks to the smartphone camera, of the inherent power of social media to short-circuit established channels of 'old' media, I suggest, remain at what Deleuze (1988, p. 76) calls the molar level of macro-phenomena. By way of a descent into the molecular and the microphysical, I now examine the cases of two contrasting interviews conducted by two different social media pages with members of the fishing hamlets affected by the violent crackdown of the protest.

## 4. Aftermath 1—Dalit Camera's Youtube Page

Dalit Camera, a YouTube channel focusing on caste-based issues of the Dalit and other oppressed communities, shot a series of videos, titled 'Chennai Police Brutality', for which they interviewed people living in the fishing villages near Marina Beach. The videos feature mostly women in the area who describe the entry of riot police into the area, tear gas shelling, the burning of the Nadukkuppam fish market, the vulgar language by which the police addressed them. They spoke of their sympathy for the protestors, 'defenceless kids', who ran into their areas crying that the police were attacking them. The women did not mince their words. They wanted the men of the area and the boys who had been arrested to be released. They wanted compensation for their damaged property. They wanted action to be taken on the policemen. 'Aren't we Tamils too? They set upon us with such hateful force! The students aren't anti-socials or terrorists, it is the police who are terrorists. They burnt our entire market. They burnt our vehicles'.

In the first video of the series, a man from Rutherpuram declares that the state government let the police off their leash.

> Every party, every policy, centre, state, is against us... Look at us here. We live in the middle of garbage. We don't have clean water. No fresh vegetables. We have nothing and they tell us we are Tamils, that we are Indians. India itself is two today. In every town, every village in Tamil Nadu, there is the Ur [town] and then there is the Cheri or Kuppam [referred to in colloquial Tamil speech by the English word 'colony' to refer to settlements][3]. Give us our share. We want two of everything, if there is to be an Ur and a Kuppam, we want two Chief Ministers, two MLAs, give us our rightful share of the land. Our people haven't even seen Jallikattu on TV. We stood with the protesters because it was a fight for rights. The police quarters is right here. How long will it take for us to scale the wall and burn it down? We don't do it because we do not practice violence. My message to the government is: Do not turn those who are not violent, onto the path of violence.

Throughout this talk, the interviewer, Greeshma Rai, an activist and contributor on the Dalit Camera channel, does not stop him. At one point, a voice can be heard prompting him to speak about the police and their use of force and that they must be investigated and condemned for their action. The man, however, interprets the prompt in his own way and speaks of scaling the walls to the police quarters. The videographer does not censor

him. The tactics of the videographer allowed for the person addressing the camera to speak and, in this speech, constitute himself as a political subject who, as target of violence, demanded political compensation. The interviewee is allowed a space for expression without interruption or over-coding.

A couple of days after a series of videos were uploaded, Dalit Camera's YouTube channel was terminated for 'Copyright infringement'. Video creators from Dalit Camera composed and forwarded a message to their friends and chat groups, requesting people to mail Google with the URL link to the channel, with a note declaring that 'the channel uploaded original activist content representing critical voices of marginalized groups'. It was back online a couple of days later, but some of the videos stayed blocked for a while longer. The number of views on these videos is of the order of a few thousand.

## 5. Aftermath 2—Video on the Chennai Memes Facebook Page

On February the first, a team from Chennai Memes (the same social media group that organized the walkathon to save Jallikattu) visited Nadukkuppam, the neighbourhood with the burnt-down fish market. A group of three women were interviewed[4]. The interviewer asks them to collect a 'list of things, essentials' that the people of Nadukkuppam require. In a remarkable contrast to the videos shot in the same vicinity by the Dalit Camera channel, these women are docile and speak the transactional language of grains of rice, dal, blankets for the night and fresh sarees for the morrow. When the interviewer asks them if there continues to be trouble from the police, they say that there has been no further trouble. The interviewer proceeds to ask them if they have received help already, to which the women reply that different political personalities had visited the area, had promised and had given them rice and funds for repair, and that the government had promised a temporary fishing market would be built soon.

The next day, the minister of fisheries arrived in the area to inaugurate the fish market. A team of policemen are also there. With speakers blaring music of the nadaswaram and mridangam, fixtures in the Carnatic music characteristic of the Brahmin areas of Mylapore, the minister poses with women in the area for photographs. Sri Ganesh from Chennai Memes, a mover in the peripheries of politics, in addition to organizing the Jallikattu walkathon, covers this event. Before the video ends, he faces the camera and tells viewers that Chennai Memes have been spending the money donated to them wisely.

Later that day, an interviewer from Chennai Memes speaks to two women seated at the newly inaugurated market. He then lists various items that the Chennai Memes team has provided them with thus far, presented as something of a transparent audit to those who donated the items to the Chennai Memes social media team. The interviewer then asks them what else they would need to restart their lives. The women explain that they obtain fish from the market taking out a loan from money lenders. The fish they sell are expensive and, hence, cost a lot too. They do not invest their own money in buying the fish since they do not have such large sums saved up. The burning of the fish market left them with no fish to sell and with an outstanding loan to pay. They tell the interviewer to spread word to people on their Facebook page, to support them as customers, buying from them at the market. The interviewer tells them that the Chennai Memes team will try to do whatever they can for them, that they would try to prompt the government to address their grievances and that they would refer their shop to their friends and Facebook subscribers. At this point, the women reply that no relief had been offered to them from the government and that they never extended their hands and asked anyone for anything. Agitated, one of the women stands up, her hands gesturing with emotion, and says that whatever came for them thus far by way of aid had already been appropriated by certain parties. She is about to say more, but, at this point, the interviewer cuts her off and tells the women that the government has promised to transfer money into their bank accounts. He continues talking and leaves them no room to say anything further. He ends the interview, listing once again the things that Chennai Memes had arranged for them, the bags of rice, the packets of dal, that they will do whatever they can for them, and then thanks them for

the interview. In contrast to the Dalit Camera video, the tactics of the interviewer allow for the women to produce themselves as victims lacking political agency, who need the aid of the Chennai Memes team, and viewers of this channel, to restart their lives. As soon as they move beyond the language of relief and aid, treading the terrain of political subjectivity, the interviewer interrupts them.

Within this framing, the following story is told: the Tamils who stood as one during the protest achieved their aims. Towards the end, an unnameable misfortune befalls the fisherfolk near the beach. But the Tamils have come together once again to support them. What is glaring here is the docility of the interviewees, whose political voices are muted. They are traumatized subjects who must not speak too much truth but who should just make do with the transactional offering of grain and cloth.

## 6. Conclusions

Foucault (1995, p. 27) suggests that power at the level of the micropolitical is 'not acquired once and for all by a new control of the apparatuses nor by a new functioning or a destruction of the institutions'. Rather, the micropolitical level of power, for Foucault, is located in 'the effects that it induces on the entire network in which it is caught up'. I have argued that claims of social media's trigger-potential, of their power to sousveil, and of their promise of a new organization of power relations miss the micropolitical. To speak of social media's "affordances", I argue, treats power as 'acquired once and for all' and objectified within technological interfaces. I have demonstrated through the video interviews analysed above how the differential construction of relations of power allows specific forms of subjectivity to appear and, as in the case of the Chennai Memes video, can constitute a limit on what can be said. Both the videos are shared on social media platforms, but the relations of power that cut across them are vastly different. The Dalit Camera videos that allow for the uninterrupted formation of political subjectivity garner barely a thousand views, and the Chennai Memes videos frame the people of the fishing hamlets as subjects in need of relief and aid, as if they were afflicted by a flood, and, in this, we find a tactic that resonates with the strategy of the state.

In contrast to the rousing images and speeches from the Jallikattu protest, these videos narrated stories about themselves that the Tamils could barely tell other Tamils, stories about themselves that other Tamils could barely stand listening to, without turning away.

**Funding:** This research received no external funding.

**Institutional Review Board Statement:** Not applicable.

**Informed Consent Statement:** Not applicable.

**Data Availability Statement:** Not applicable.

**Conflicts of Interest:** The author declares no conflict of interest.

## Notes

[1]　See the link for the post. Available online: https://www.facebook.com/events/1559079037440854/ (accessed on 6 October 2022).

[2]　See link below. Available online: https://www.facebook.com/events/1559079037440854/ (accessed on 15 December 2020).

[3]　The Kuppams are segregated 'colonies' for the scheduled castes, the administrative category describing Dalits, and are usually situated at a distance from the 'Ur', the village area where the upper castes resided. See link below. Translations are mine. Available online: https://www.youtube.com/watch?v=Vq-V2GXh2pc (accessed on 6 October 2022).

[4]　See link below. https://www.facebook.com/watch/live/?v=743349669164500&ref=watch_permalink (accessed on 6 October 2022).

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
