# Peer review of "Power and Subjectification at the Edge of Social Media Interfaces in the Aftermath of the Jallikattu Protest"

_humanities, doi:10.3390/h12040082_

Round 1

Reviewer 1 Report

The basic issue of this otherwise very interesting submission is its lack of engagement with theory in an original and problematising fashion. Other than engaging with Foucault (though only marginally), I do not detect a comprehensive engagement with theory. To my mind, the author of this paper ought to theoretically contextualise his/her understanding of power and, what's more, state power. Indeed -- and although I understand that it is challenging to contextualise the particulars of Jallikattu for a wider audience -- it seems that all the chapters read like an interesting piece in The New York Times rather than a research paper that sets out to conceptually contextualise this particular phenomenon within a more philosophically-driven desire to explore the ways in which state power clashes with popular will. I am not familiar with statecraft in India, but to the extent that, I gather, India has set up its organisation along colonial/western lines (PM, government, Supreme Court etc), the author ought to offer a more conceptually problematised (and problematising) understanding of state power and resistance. Alongside Foucault on power, I would suggest Hobbes, Gramsci, perhaps Bentham and Foucault, Agamben, Schmitt and so on. In terms of resistance, I would suggest Eagleton, Couzens Hoy, Mouffe, and/or Howard Caygill. Of particular interest here would also be the ways in which central state-power is administered on the, now, most populous country on earth. 

Author Response

Many thanks for your comments and suggestions. Please see the attached file for my responses.  

Reviewer 2 Report

The article is about an important topic and in principle the review Cracks in the screen – the microphysics of power and the fracture of resistance at the edge of social media interfaces is a good idea, also performed as it comes to literature search in an adequate way. 

1- the title is too limited and does not reflect the real content

2- most of the pictures do not add any value, and are also unreadable in current form 

3- The manuscript should be written in a clear and concise language that is accessible to a wide range of readers. Technical jargon and overly complex language should be avoided where possible.

4-  Relevant and up-to-date references: The manuscript should include relevant and up-to-date references to support the arguments and claims made in the Cracks in the screen . The references should be from reputable sources and should cover a range of perspectives on the topic. At its core, social media is a technology that allows individuals to connect and communicate with each other on a massive scale. While this can be a powerful tool for organizing, mobilizing, and challenging power, it can also be used to reinforce existing power structures and hierarchies at manuscript .

5- The manuscript should present a rigorous analysis of the topic, drawing on a range of theoretical and empirical perspectives where appropriate. The analysis should be well-structured and should clearly present the key arguments and findings.

6- One of the key ways in which power is exercised on social media is through the design of the interfaces themselves. Social media platforms are designed to encourage certain types of behavior, and these designs can have profound effects on how users interact with each other and with the platform itself.  so the manuscript should make an original contribution to the field, either by presenting new empirical data or by offering a new theoretical perspective on the topic. The contribution should be clearly articulated and should be of interest to scholars in the field.

7- The argument you are making is that the claim that new media has a lesser or greater potential to resist colonization cannot be made a priori, and that it is important to analyze specific discursive sites, such as the popular Facebook group and the YouTube channel centered around Dalit issues, in order to understand the ways in which new media can be used to resist colonization.This is an important argument, as it challenges the assumption that new media is inherently resistant or susceptible to colonization. Instead, it suggests that the potential for resistance or colonization is contingent on a variety of factors, including the specific discursive sites in which new media is used, the social and cultural contexts in which these sites operate, and the ways in which users engage with these platforms. By focusing on specific discursive sites, you are able to provide a more nuanced analysis of the potential for resistance and colonization within new media. This approach allows for a more fine-grained understanding of the ways in which new media can be used to challenge dominant discourses and power structures, as well as the ways in which these platforms can be co-opted by those in power to reinforce existing hierarchies and marginalize marginalized groups.

8- The research questions and objectives should be clearly stated in the manuscript. The manuscript should explain why the research is important, what gaps in knowledge it aims to fill, and how it will contribute to the field.

9- the structure that you have small sections about each read article with reference (section 7.8) is not usable in my view, at least against all principles of typical scientific writing

10- The abstract lacks coherence and the method is not figured properly. Would suggest using shorter sentences.

Author Response

Thank you for your critical comments and suggestions. I have made changes to the manuscript. Please see the attached file for point-wise responses. 

Reviewer 3 Report

This is an important piece that contributes to the broader field of knowledge about the contested role of social media in civic unrest.

There are, however, several areas that are missing references. Without them, this article at times comes across as mere conjecture. References are missing at lines: 8; 29; 46; 154; 172; 293; 301.

Ravindran (line 52) does not appear in the bibliography. Cody's name (line 59) is the only ref that uses a first name - please be consistent. It would also be nice to have the links - in text - to the examples the author is analysing.

The abstract tells us social media is being compared with broadcast media. I am wondering why just broadcast, and not print?  

Overall, this is a beautifully-told narrative, and I have learned something new by reviewing it. Thank you

Author Response

Thank you for reading the article with care, and thank you also for your comments. I have actioned almost all the changes you suggested. Please see the attached file for a more detailed response to your comments. 

Round 2

Reviewer 1 Report

I am satisfied by the author's comments in response to my initial review. 

Reviewer 2 Report

The manuscript can be published

The manuscript can be published